# Cerebral cortex activation and functional connectivity during low-load resistance training with blood flow restriction: An fNIRS study

Binbin Jia[1,2☯], Chennan Lv[3☯], Danyang Li[1,2], Wangang Lv[3]*

**1** School of Sports Training, Wuhan Sports University, Wuhan, China, **2** School of Physical Education, Wuhan Sports University, Wuhan, China, **3** Center of Strength and Conditioning, Wuhan Sports University, Wuhan, China

☯ These authors contributed equally to this work.
* 937757336@qq.com

**Data Availability Statement:** The raw data of fNIRS, R code, and statistical results from JASP were unloaded in figshare (DOI:10.6084/m9.figshare.25560594).

## Abstract

Despite accumulating evidence that blood flow restriction (BFR) training promotes muscle hypertrophy and strength gain, the underlying neurophysiological mechanisms have rarely been explored. The primary goal of this study is to investigate characteristics of cerebral cortex activity during BFR training under different pressure intensities. 24 males participated in 30% 1RM squat exercise, changes in oxygenated hemoglobin concentration (HbO) in the primary motor cortex (M1), pre-motor cortex (PMC), supplementary motor area (SMA), and dorsolateral prefrontal cortex (DLPFC), were measured by fNIRS. The results showed that HbO increased from 0 mmHg (non-BFR) to 250 mmHg but dropped sharply under 350 mmHg pressure intensity. In addition, HbO and functional connectivity were higher in M1 and PMC-SMA than in DLPFC. Moreover, the significant interaction effect between pressure intensity and ROI for HbO revealed that the regulation of cerebral cortex during BFR training was more pronounced in M1 and PMC-SMA than in DLPFC. In conclusion, low-load resistance training with BFR triggers acute responses in the cerebral cortex, and moderate pressure intensity achieves optimal neural benefits in enhancing cortical activation. M1 and PMC-SMA play crucial roles during BFR training through activation and functional connectivity regulation.

## Introduction

Blood flow restriction (BFR) training typically involves using pneumatic cuffs placed around the limbs to limit blood inflow into the muscles during exercise [1]. It has been proven to promote muscle hypertrophy and strength gains among individuals with varying athletic abilities and low exercise loads [2–4]. While BFR has been combined with various types of exercise, research indicates that the most substantial muscular gains come with resistance training (RT) under 20%-40% of the 1 repetition maximum (1RM) or maximum voluntary contraction

**Funding:** This work was supported by the Scientific Research Center at Wuhan sports university, China under project number 2022J03.

**Competing interests:** The authors have declared that no competing interests exist.

(MVC) [5]. With the increasing popularity of BFR in the training domain, many researchers have started to investigate its potential mechanisms, such as metabolic stress, cellular swelling, hormone regulation, and other mechanisms at the cellular and molecular levels [6–8]. Surprisingly, even though neural regulation has significantly contributed to muscle hypertrophy and strength gain [9–12], studies concerning BFR training in this area remain limited. However, there is already evidence suggesting the regulation of neural systems during BFR training. For instance, researchers have found that BFR training affected the electromyography signal, which supports the acute response of neuro-muscular [13,14]. In addition, studies have shown that muscle hypertrophy and strength gain can transfer from muscles exposed to BFR to muscles not exposed to BFR [15,16]. Moreover, Sugimoto et al.[17] found that combining BFR with walking enhanced participants' performance in cognition tasks. However, studies employing electromyography have limitations when exploring the neural regulation process [18], transfer effect in muscle hypertrophy and strength gain, as well as cognition enhancement with BFR training only provide indirect evidence. To gain a deeper understanding of the neurophysiological mechanisms associated with BFR training, it is crucial to provide robust evidence for the characteristics of activity within the central nervous system (CNS), such as activation and functional connectivity (FC) of the cerebral cortex. Those indices not only serve as excellent windows for exploring the response pattern of the cerebral cortex, but have also been confirmed to undergo adaptive changes with resistance training[19].

Current evidence concerning the cortical response induced by BFR training is limited. A previous study by Morita et al. [20] reported increased activation in the prefrontal cortex during BFR training compared with non-BFR. Similarly, Brandner et al. [21] assessed changes in motor-evoked potentials (MEPs) by transcranial magnetic stimulation, revealing higher MEP amplitudes following BFR training. However, these studies suffered from a notably small sample size, and using MEPs to measure cortical excitability carries inherent limitations [22]. Additionally, the dose-response effect of pressure intensity, a key variable influencing the effectiveness of BFR training [2,5,23], on cortical activity has not been examined yet. As a result, the evidence supporting cortical regulation during BFR training remains weak and incomplete. Furthermore, there is a possibility that the increases in cortical activation from the prior study [20] are passive consequences of altered blood distribution and increased cerebral blood flow during BFR training, rather than active regulation of the CNS. However, this hypothesis has yet to be tested.

Functional near-infrared spectroscopy (fNIRS) has been widely used to examine brain activity. It is well-suited for monitoring cortical response during exercise scenarios [24]. This technique enables us to evaluate characteristics of cerebral cortex activity throughout BFR training by examining the concentration changes of oxygenated hemoglobin (HbO) and deoxygenated hemoglobin (HHb). It's important to mention that we chose HbO as the primary indicator for assessing cortical activation and functional connectivity in this study. This is due to the HbO's superior signal-to-noise ratio, reliability, heightened sensitivity to cortical blood flow changes, and more significant contribution to overall oxygen signal compared with HHb [25–28]. Additionally, the ROIs we focused on in this research include primary motor cortex (M1), pre-motor cortex (PMC), supplementary motor area (SMA), and dorsolateral prefrontal cortex (DLPFC). These regions not only play significant roles in motor planning and execution [29–31] but are also crucial in facilitating the induction of CNS adaptations resulting from exercise [10,32].

Therefore, the primary purpose of this study is to provide stable and comprehensive evidence about the cortical response during BFR training. Specifically, we intend to combine the 30% 1RM squat exercise with BFR under different pressure intensities (150 mmHg, 250 mmHg, 350 mmHg, and 0 mmHg or non-BFR as a control condition) to investigate the

cortical activation and FC in M1, PMC-SMA, DLPFC via fNIRS. The first hypothesis is that pressure intensity affects cortical response. Furthermore, we infer activation and FC strength would enhance with pressure intensity to improve muscle force output [19, 33,34]. This is because the increasing metabolic stress with BFR restricts the capacity of muscle [6,34], and the CNS can compensate for the muscle force loss under BFR by enhancing the recruitment of motor units, and improving the frequency of neural impulse discharge[35], resulting in higher activation. Additionally, considering the changes in cortical HbO during BFR training may result from altered blood distribution and increased cerebral blood flow [36,37], rather than active regulation of the CNS, we propose the second hypothesis that the influence of pressure intensity on cortical activation is moderated by the regions of interest (ROI). This is based on the assumption that if the ROIs in our study play an active role in the regulation of CNS during BFR training, we will detect an interaction effect between pressure intensity and ROIs regarding cortical activation. Conversely, if changes in cortical activation result from variations in cerebral blood flow during BFR training, the regulatory effects of pressure intensity across different ROIs should be consistent.

## Methods

### Participants

This study enrolled 24 male participants (age, 20.08 ± 0.93 years; height, 179 ± 5 cm; weight, 73.63 ± 10.53 kg; 1RM, 133.33 ± 15.37 kg). The sample size determination was based on prior-power analysis in G*Power and MPower [38,39]. More details about the prior power analysis can be found in the S1 Appendix. To reduce the potential risk of injury during resistance training (RT), all participants had at least 1 year of squat training experience (3.52 ± 1.25 years). Exclusion criteria included neurological or psychological disorders (Depression, Autism, Mania, Schizophrenia, Epilepsy, Stroke, etc.), the use of medications affecting the CNS, consumption of caffeine or alcohol within 24 hours before the experiment, acute or chronic exercise-related injuries, as well as cardiovascular diseases. The recruitment period started on June 10, 2023, and ended on November 30, 2023. Written informed consent was obtained from all participants before the study, and ethical approval for this experiment was granted by the Ethics Committee of Wuhan Sports University (Approval No. 2023050).

### Experiment material and task

The tools utilized in this study included a squat rack, a near-infrared imaging system (NIRx-sport2, NIRx Medizintechnik GmbH, Berlin, Germany), and pneumatic cuffs with a width of 7cm (B-Strong, USA). The task was conducted with E-prime 2.0 (Psychology Software Tools, Pittsburgh, USA). All experiments took place within a laboratory isolated from external light and noise. Specifically, participants were initially presented with a cue indicating the preparation for a squat. Subsequently, participants performed squats following the cues displayed on the screen. After completing each squat, participants unloaded the barbell and maintained a static standing posture during an interval until the cue for the following squat preparation appeared, as illustrated in Fig 1.

The formal experiment was divided into 4 blocks, each consisting of 20 trials, with 2–3 minutes of rest inserted between blocks. BFR intensity was controlled using pneumatic cuffs at 4 pressure intensities (0 mmHg, 150 mmHg, 250 mmHg, and 350 mmHg) across these 4 blocks. The cuffs were positioned at one-third of the participants' upper thighs bilaterally during each block, with no occlusion during the rest periods between blocks, as depicted in Fig 2. Blocks were presented in a pseudo-random order to minimize the potential impact of block order on experimental outcomes. To determine the external resistance of squat in BFR training, all

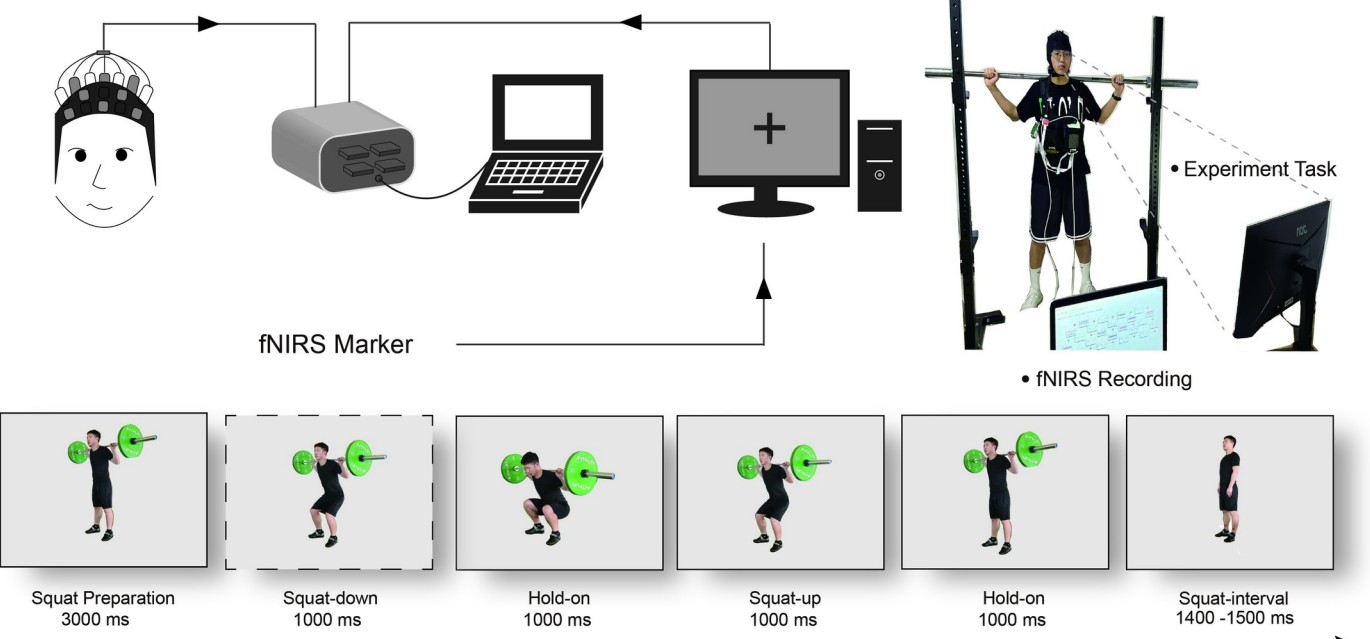

**Fig 1. The preview of the task.** The individuals in this photograph have given written informed consent (as outlined in the PLOS consent form) to publish these case details.

participants underwent a 1RM test 1–2 days before the experiment. The test started with a warm-up, followed by the squat test with the initial weight set at 70% of the participant's self-estimated 1RM. After the completion of the 1RM squat test, participants engaged in static stretching for 3–5 minutes.

## fNIRS recording

The NIRx-Sport2 (continuous wave) with wavelengths 760 and 850 nm was used to record changes in cortical HbO at a sampling rate of 10.2 Hz. This system has 8 light sources and 7 detectors, which form 22 channels (as illustrated in Fig 3). These channels mainly covered M1, PMC-SMA, and DLPFC. The placement of the light sources and detectors was determined by fNIRS optodes' Location Decider [40]. The brain atlas referred to the Brodmann Brain Regions and the coordinates for the light sources and detectors were based on the 10–10 international system.

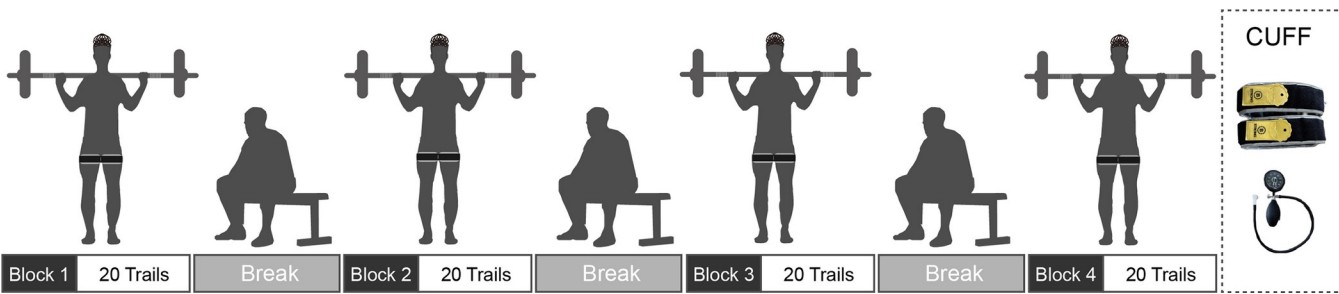

**Fig 2. Overview of testing procedures.**

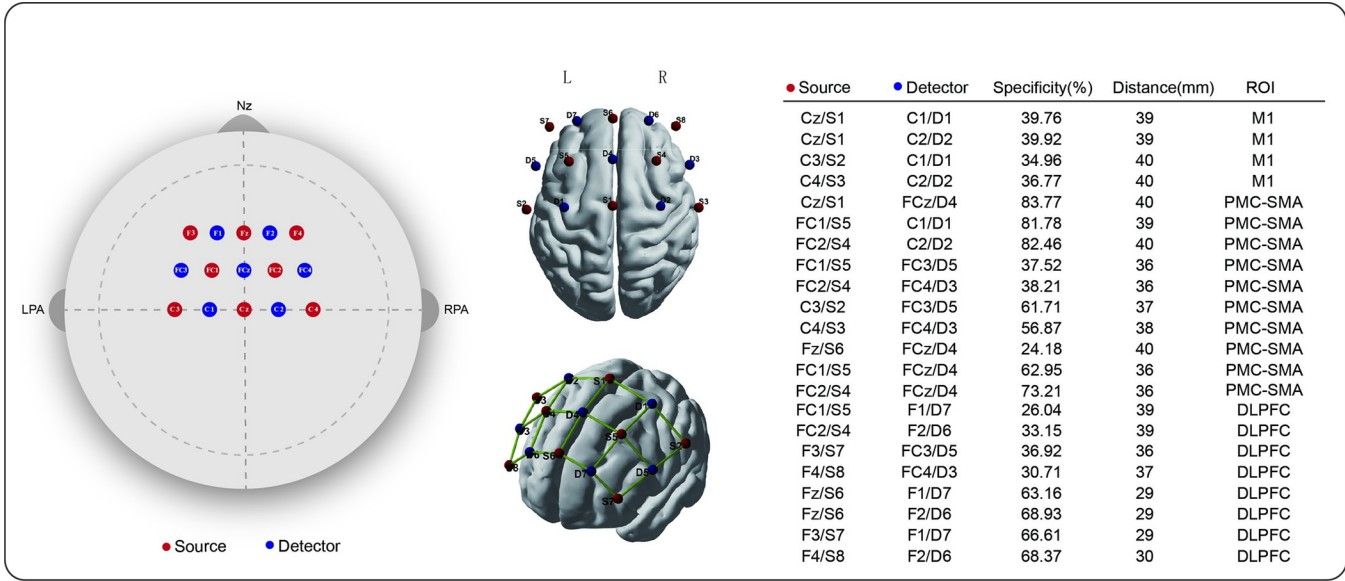

**Fig 3. The layout and set-up information of fNIRS channels.** BrainnetViewer visualized the channel layout with the smoothed Colin brain template [41]. The coordinates of the nodes corresponded to the positions of the light sources (red) and detectors (blue), while the edges represented the 22 channels. The term 'specificity' refers to the representativeness of each channel for its corresponding brain area based on its anatomical location.

## Data analysis and statistics

The fNIRS data were processed using HOMER3 [42]. The data processing workflow is illustrated in Fig 4, and more details can be found in the S2 Appendix. The $HbO_{mean}$ and $HbO_{max}$ were extracted from temporal changes of HbO during BFR training under different conditions for further statistics. The FC index represents Pearson's correlation of HbO between channels from 2s to 15s during the trial in our task, which accounts for the delay in hemodynamic response [43]. The Pearson's r values were then translated to Fisher Z (Z = 0.5 * ln((1+r)/(1-r)). Afterward, the fNIRS data underwent pre-processing in R (https://www.r-project.org/) for visualization and were then input into JASP (https://jasp-stats.org/, version 0.16.4) for statistical inference. The raw data of fNIRS, R code, and statistical results from JASP were unloaded in figshare (DOI:10.6084/m9.figshare.25560594).

Repeated measures analysis of variance (RMANOVA) was employed to test our hypotheses. The significance level was set at 0.05. For both main effects and interaction, we provided the F-value, p-value, and effect size partial eta square ($\eta_p^2$). In cases where sphericity was violated, Greenhouse-Geisser-corrected statistics were reported. Multiple comparisons were conducted using a paired-sample t-test. Statistical information, including t-value, p-value, and effect size

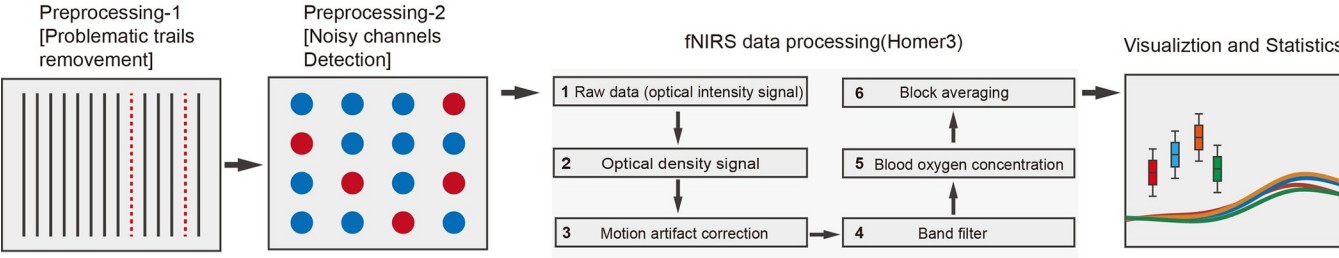

**Fig 4. fNIRS data processing workflow.**

Cohen'd with its 95 confidence interval (95%CI), was provided. It is important to note that due to missing data for some participants in specific experimental conditions, multivariate imputation was performed using the MICE package [44] to maintain data balance and predetermined statistical power of this 2 within-factors (pressure intensity and ROI) repeated measures design. Details regarding the imputation of specific variables will be shown in the Result section.

## Results

### HbO$_{mean}$ under different pressure intensities during BFR training

RMANOVA for HbO$_{mean}$ was conducted with a data imputation rate of 1.36% (15/1104). This analysis revealed significant main effects of pressure intensity ($F(3,66)$ = 9.55, $\eta_p^2$ = 0.3, p<0.001) and ROI ($F(2,44)$ = 12.59, $\eta_p^2$ = 0.36, p<0.001). Moreover, a significant interaction effect between pressure intensity and ROI was detected ($F(6,132)$ = 2.32, $\eta_p^2$ = 0.1, p = 0.04). Subsequently, a simple main effects analysis demonstrated that the regulatory effect of pressure intensity on HbO$_{mean}$ was more pronounced in M1 ($F(3,66)$ = 9.67, $\eta_p^2$ = 0.31, p<0.01) and PMC-SMA ($F(3,66)$ = 9.00, $\eta_p^2$ = 0.29, p<0.01) compared to DLPFC ($F(3,66)$ = 2.24, $\eta_p^2$ = 0.09, p = 0.09). Between condition comparisons were illustrated in Fig 5, more detailed statistical results can be found in Table 1–3.

### HbO$_{max}$ under different pressure intensities during BFR training

RMANOVA for HbO$_{max}$ was also conducted with a data imputation rate of 1.36% (15/1104). This analysis revealed significant main effects of pressure intensity ($F(3,66)$ = 10.22, $\eta_p^2$ = 0.2, p<0.001) and ROI ($F(2,44)$ = 19.34, $\eta_p^2$ = 0.47, p<0.001). Moreover, a significant interaction effect between pressure intensity and ROI was also detected ($F(6,132)$ = 3.4, $\eta_p^2$ = 0.13, p = 0.03). Subsequently, a simple main effects analysis demonstrated that the regulatory effect of pressure intensity on HbO$_{max}$ was more pronounced in M1 ($F(3,66)$ = 13.56, $\eta_p^2$ = 0.38, p<0.001) and PMC-SMA ($F(3,66)$ = 10.16, $\eta_p^2$ = 0.32, p<0.001) compared to DLPFC ($F(3,66)$ = 1.34, $\eta_p^2$ = 0.06, p = 0.27). Between condition comparisons were illustrated in Fig 6, more detailed statistical results can be found in Table 4–6.

### FC under different pressure intensities during BFR training

The FC between fNIRS channels during BFR training was presented in Fig 7. RMANOVA for FC was conducted with a data imputation rate of 3.99% (88/2208). The main effect of ROI was significant ($F(5,110)$ = 16.99, $\eta_p^2$ = 0.43, p<0.01, between condition comparisons are illustrated in Fig 8, more detailed statistical results can be found in Table 7. Meanwhile, the main effect of pressure intensity did not reach statistical significance ($F(3,66)$ = 2.29, $\eta_p^2$ = 0.09, p = 0.11), as well as the interaction effect between pressure intensity and ROI ($F(15,330)$ = 0.67, $\eta_p^2$ = 0.03, p = 0.67).

## Discussion

This research confirms the occurrence of cortical regulation during BFR training. To our knowledge, this study was the first to investigate the cortical activation and FC pattern during BFR training under different pressure intensities. Initially, our results demonstrated that the pressure intensity during BFR training affects cortical activation. Specifically, we found an increase in HbO from 0 mmHg to 250 mmHg, which is consistent with previous research [20].

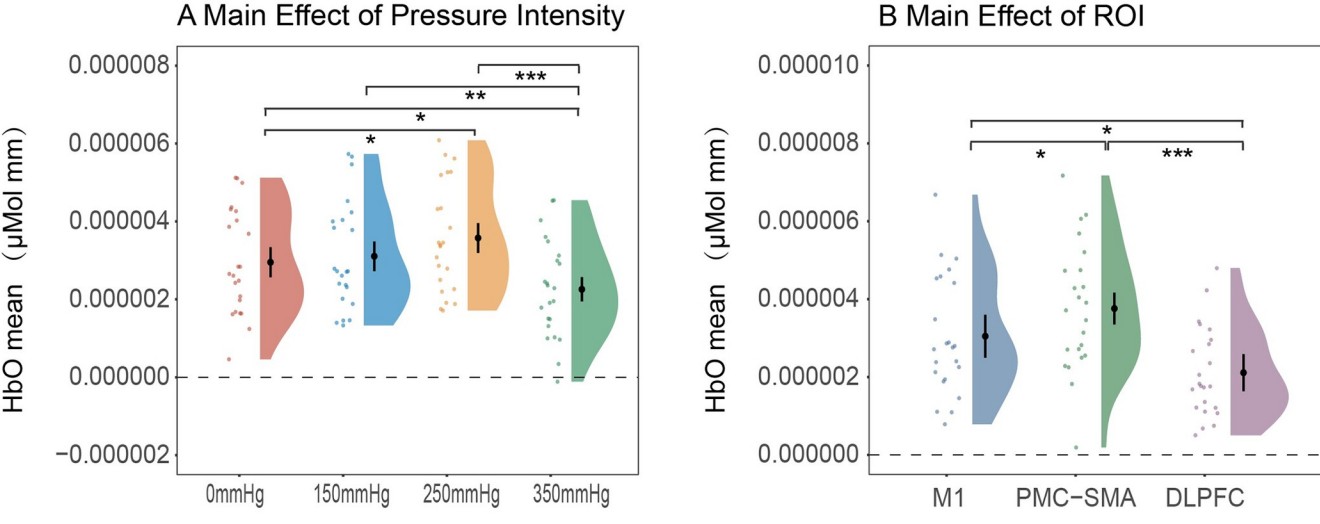

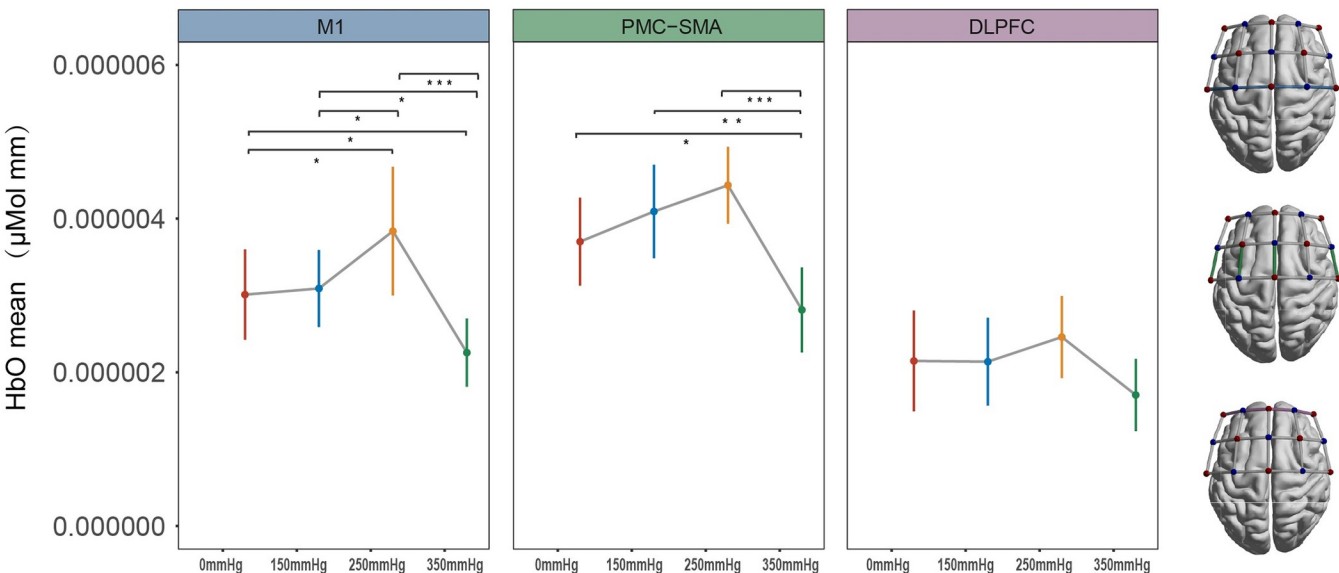

**Fig 5. Statistical results of HbO$_{mean}$ under different pressure intensities and ROI.** The error bars represent mean ± 95% CI. * p < .05, ** p < .01, *** p < .001.

This phenomenon can be interpreted as a compensatory response by the CNS to improve muscle force. It occurs as a result of heightened metabolic stress during BFR training, which includes decreased oxygen saturation and the accumulation of metabolic waste products like blood lactate, carbon dioxide, and hydrogen ions, which limit muscle capacity [6,45]. In this context, the CNS enhances muscle force output by recruiting large motor units and a higher neural impulse firing rate [35,46]. Consequently, there is an elevation in cortical HbO, ensuring cerebral energy supply and subsequently leading to increased cortical activity during BFR training. Furthermore, this heightened activity has been consistently associated with elevated force output [34,47,48].

**Table 1. Post Hoc Comparisons—Pressure Intensity (HbO$_{mean}$).**

| Comparison | | t | d | 95% CI | | p$_{holm}$ | Sig |
|---|---|---|---|---|---|---|---|
| | | | | Lower | Upper | | |
| 0mmHg | 150mmHg | -0.62 | -0.09 | -0.50 | 0.32 | 0.54 | |
| | 250mmHg | -2.49 | -0.36 | -0.80 | 0.07 | 0.04 | * |
| | 350mmHg | 2.78 | 0.41 | -0.03 | 0.85 | 0.03 | * |
| 150mmHg | 250mmHg | -1.88 | -0.27 | -0.70 | 0.15 | 0.13 | |
| | 350mmHg | 3.40 | 0.50 | 0.04 | 0.95 | 0.006 | ** |
| 250mmHg | 350mmHg | 5.28 | 0.77 | 0.25 | 1.29 | <0.001 | *** |

The p-values were corrected by the Holm method, and d (Cohen's d) along with its 95%CI were corrected by the Bonferroni method.

**Table 2. Post Hoc Comparisons—ROI (HbO$_{mean}$).**

| Comparison | | t | d | 95% CI | | p$_{holm}$ | Sig |
|---|---|---|---|---|---|---|---|
| | | | | Lower | Upper | | |
| M1 | PMC | -2.16 | -0.42 | -0.92 | 0.09 | 0.04 | * |
| | DLPFC | 2.84 | 0.55 | 0.03 | 1.07 | 0.01 | * |
| PMC | DLPFC | 5 | 0.96 | 0.37 | 1.56 | <0.001 | *** |

Interestingly, the increase of HbO we observed during BFR training is similar to the results obtained from RT with heavier loads [34]. This pattern of cortical activity reflects a shared mechanism in regulating muscle force output by the CNS. Furthermore, it elucidates why low-load RT with BFR can induce muscle hypertrophy, and strength gains comparable to moderate or high-load RT from the perspective of cortical response. However, we noted that the overall

**Table 3. Multiple Comparisons in Different ROI (HbO$_{mean}$).**

| ROI | Comparison | | t | d | 95% CI | | p$_{holm}$ | Sig |
|---|---|---|---|---|---|---|---|---|
| | | | | | Lower | Upper | | |
| M1 | 0mmHg | 150mmHg | -0.27 | -0.05 | -0.51 | 0.42 | 0.79 | |
| | | 250mmHg | -2.81 | -0.47 | -0.98 | 0.04 | 0.03 | * |
| | | 350mmHg | 2.57 | 0.43 | -0.07 | 0.93 | 0.04 | * |
| | 150mmHg | 250mmHg | -2.54 | -0.42 | -0.92 | 0.07 | 0.04 | * |
| | | 350mmHg | 2.84 | 0.48 | -0.03 | 0.98 | 0.03 | * |
| | 250mmHg | 350mmHg | 5.38 | 0.90 | 0.30 | 1.50 | <0.001 | *** |
| PMC-SMA | 0mmHg | 150mmHg | -1.19 | -0.21 | -0.70 | 0.28 | 0.48 | |
| | | 250mmHg | -2.23 | -0.39 | -0.90 | 0.12 | 0.09 | |
| | | 350mmHg | 2.69 | 0.47 | -0.05 | 0.99 | 0.04 | * |
| | 150mmHg | 250mmHg | -1.04 | -0.18 | -0.67 | 0.31 | 0.48 | |
| | | 350mmHg | 3.89 | 0.67 | 0.12 | 1.23 | 0.001 | ** |
| | 250mmHg | 350mmHg | 4.92 | 0.85 | 0.25 | 1.46 | <0.001 | *** |
| DLPFC | 0mmHg | 150mmHg | 0.03 | 0.01 | -0.55 | 0.57 | 0.97 | |
| | | 250mmHg | -1.06 | -0.22 | -0.78 | 0.35 | 0.83 | |
| | | 350mmHg | 1.51 | 0.31 | -0.27 | 0.88 | 0.67 | |
| | 150mmHg | 250mmHg | -1.10 | -0.22 | -0.79 | 0.34 | 0.83 | |
| | | 350mmHg | 1.48 | 0.30 | -0.27 | 0.88 | 0.67 | |
| | 250mmHg | 350mmHg | 2.58 | 0.52 | -0.08 | 1.13 | 0.07 | |

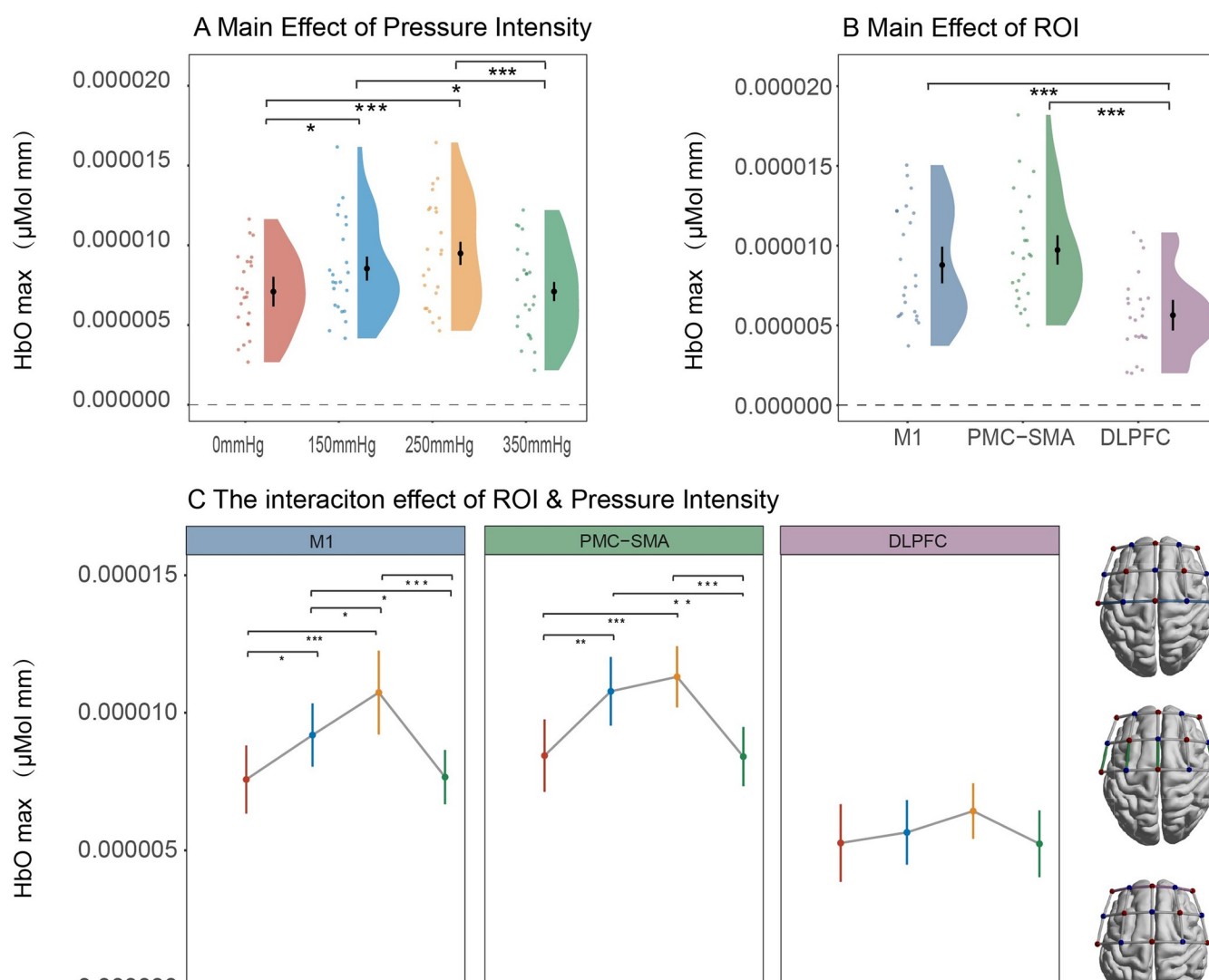

**Fig 6. Statistical results of HbO$_{max}$ under different pressure intensities and ROI.**

changes in HbO levels under different pressure intensities were lower than training with various external loads. This can be attributed to the reduction in cerebral blood flow caused by BFR, which may limit the CNS's capacity to enhance muscle force output fully. This inference

**Table 4. Post Hoc Comparisons—Pressure Intensity (HbO$_{max}$).**

| Comparison | | t | d | 95% CI | | p$_{holm}$ | Sig |
|---|---|---|---|---|---|---|---|
| | | | | Lower | Upper | | |
| 0mmHg | 150mmHg | -2.79 | -0.39 | -0.81 | 0.03 | 0.03 | * |
| | 250mmHg | -4.62 | -0.65 | -1.12 | -0.17 | <0.001 | *** |
| | 350mmHg | -0.01 | 0.00 | -0.39 | 0.39 | 0.99 | |
| 150mmHg | 250mmHg | -1.83 | -0.26 | -0.66 | 0.15 | 0.14 | |
| | 350mmHg | 2.77 | 0.39 | -0.03 | 0.81 | 0.03 | * |
| 250mmHg | 350mmHg | 4.60 | 0.65 | 0.17 | 1.12 | <0.001 | *** |

**Table 5. Post Hoc Comparisons—ROI (HbO$_{max}$).**

|                 |          |          |          | 95% CI   |          |                |       |
| --------------- | -------- | -------- | -------- | -------- | -------- | -------------- | ----- |
| Comparison      |          | t        | D        | Lower    | Upper    | $p_{holm}$     | Sig   |
| M1              | PMC      | -1.37    | -0.26    | -0.73    | 0.22     | 0.18           |       |
|                 | DLPFC    | 4.57     | 0.85     | 0.29     | 1.41     | <0.001         | ***   |
| PMC             | DLPFC    | 5.94     | 1.11     | 0.49     | 1.73     | <0.001         | ***   |

is also consistent with findings indicating that the increase in muscle strength after BFR with low-load RT was lower compared to high-load RT alone [49,50].

This study also found that the relationship between pressure intensity and cortical activation is not linear. Specifically, HbO declined sharply during 350 mmHg BFR training after an increase from 0 mmHg to 250 mmHg. We attribute this result to a significant reduction in cerebral blood flow induced by the high occlusion pressure. Interestingly, this non-linear pattern has also been observed in BFR studies using EMG as an index of neural response [13,14]. Considering the strong correlation between cortical activation and EMG signals [47,51], these consistent findings collectively suggest the dose-response effect between cortical activation and pressure intensity during BFR training. In addition, it is worth noting that the decrease of cortical oxygenation has also been observed in high-load exercise [52,33], and some researchers suggested this cerebral hypoxia-like phenomena may induce beneficial adaptive response of the CNS [53,54]. However, whether it supports training benefits such as muscle hypertrophy and strength gain is still under debate. Furthermore, considering the vulnerability and importance of the brain, the potential risks of cerebral hypoxia induced by high occlusion pressure need to be considered cautiously.

More importantly, the significant interaction effect in our study indicating the regulation of cortical activation by pressure intensity is moderated by ROI. Specifically, the impact of

**Table 6. Multiple Comparisons in Different ROI (HbO$_{max}$).**

|          |            |          |          |          | 95% CI   |          |              |       |
| -------- | ---------- | -------- | -------- | -------- | -------- | -------- | ------------ | ----- |
| ROI      | Comparison |          | t        | d        | Lower    | Upper    | $p_{holm}$   | Sig   |
| M1       | 0mmHg      | 150mmHg  | -2.82    | -0.42    | -0.87    | 0.03     | 0.03         | *     |
|          |            | 250mmHg  | -5.51    | -0.82    | -1.36    | -0.28    | <0.001       | ***   |
|          |            | 350mmHg  | -0.15    | -0.02    | -0.44    | 0.40     | 0.88         |       |
|          | 150mmHg    | 250mmHg  | -2.69    | -0.40    | -0.85    | 0.05     | 0.03         | *     |
|          |            | 350mmHg  | 2.66     | 0.40     | -0.05    | 0.85     | 0.03         | *     |
|          | 250mmHg    | 350mmHg  | 5.36     | 0.80     | 0.26     | 1.33     | <0.001       | ***   |
| PMC-SMA  | 0mmHg      | 150mmHg  | -3.45    | -0.59    | -1.12    | -0.05    | 0.003        | **    |
|          |            | 250mmHg  | -4.23    | -0.72    | -1.29    | -0.16    | <0.001       | ***   |
|          |            | 350mmHg  | 0.05     | 0.01     | -0.47    | 0.48     | 0.96         |       |
|          | 150mmHg    | 250mmHg  | -0.78    | -0.13    | -0.61    | 0.35     | 0.88         |       |
|          |            | 350mmHg  | 3.50     | 0.60     | 0.06     | 1.14     | 0.003        | **    |
|          | 250mmHg    | 350mmHg  | 4.28     | 0.73     | 0.17     | 1.30     | <0.001       | ***   |
| DLPFC    | 0mmHg      | 150mmHg  | -0.57    | -0.12    | -0.70    | 0.46     | 1.00         |       |
|          |            | 250mmHg  | -1.72    | -0.36    | -0.96    | 0.24     | 0.50         |       |
|          |            | 350mmHg  | 0.04     | 0.01     | -0.57    | 0.59     | 1.00         |       |
|          | 150mmHg    | 250mmHg  | -1.15    | -0.24    | -0.83    | 0.34     | 1.00         |       |
|          |            | 350mmHg  | 0.61     | 0.13     | -0.45    | 0.71     | 1.00         |       |
|          | 250mmHg    | 350mmHg  | 1.76     | 0.37     | -0.23    | 0.97     | 0.50         |       |

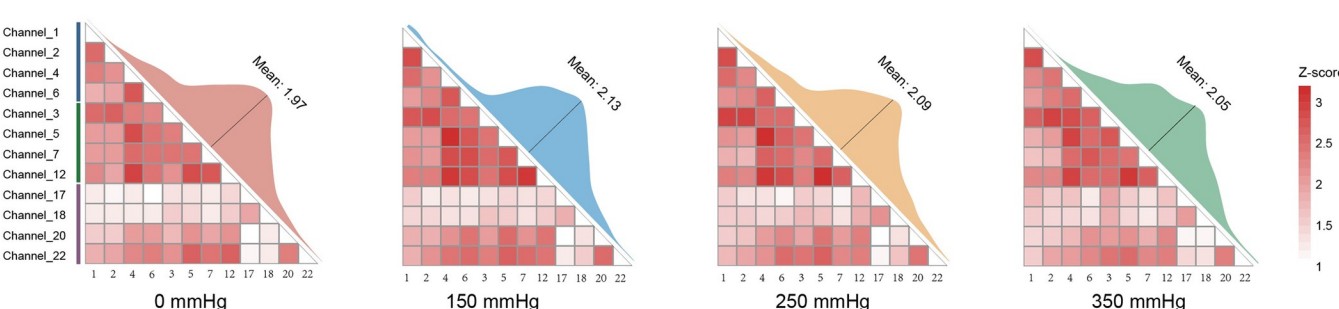

**Fig 7. FC analysis of channels under different pressure intensities and ROI.** The chord diagrams were generated based on paired t-tests comparing FC under 0 mmHg with other pressure intensities. Channels with dashed lines indicate $p < 0.05$.

pressure intensity on HbO changes was more pronounced in M1 and PMC-SMA compared to DLPFC. This finding suggests that M1 and PMC-SMA play more critical roles during BFR training. Moreover, this result also supports the active role of cerebral cortex regulation during BFR training under different pressure intensities, and the main effect of pressure intensity on HbO changes is not solely a by-product of systemic blood flow distribution variation caused by BFR [55].

For the FC index, the main effect of pressure intensity was not significant. Firstly, we attribute this to the already high connectivity during 0 mmHg or non-BFR condition, which may have limited the improvement of FC during BFR training. Secondly, the effect size in prior-power analysis was underestimated due to the limited research on FC during BFR training, which led to an insufficient sample size to detect statistically significant results. However, the raw data indicate FC in most fNIRS channels had been strengthened during BFR training, and a non-linear pattern was also observed among different pressure intensities. combining our FC results with previous findings that suggest the positive correlation between muscle force output and FC strength [19], we suggest enhancing FC is a cost-effective strategy for the CNS to increase force output during BFR training with limited muscle capacity under low pressure. Nevertheless, as the metabolic stress within muscles rises with high pressures, the CNS must employ more efficient ways like enhancement of cortical activation to increase force output,

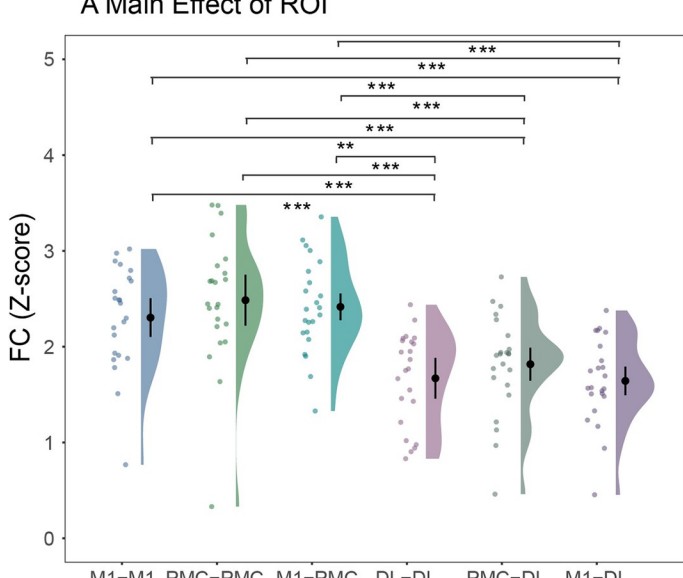
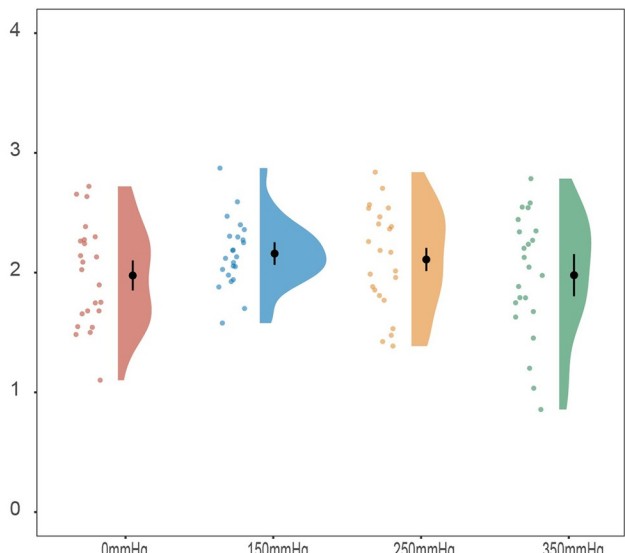

**Fig 8. Statistical results of FC under different pressure intensities and ROI.**

just like the significant increase of HbO we found from 150 mmHg to 250 mmHg. This inference also aligns with a fundamental brain characteristic, which involves the delicate balance between cost and efficiency [56].

Lastly, the higher HbO increase and stronger FC observed in M1 and PMC-SMA suggest their greater involvement compared with DLPFC during BFR training. This is consistent with previous findings highlighting the importance of PMC-SMA in spontaneous and sequential movements [57,58], as well as the dominant role of M1 in regulating muscle output parameters

**Table 7. Post Hoc Comparisons—ROI (FC).**

| Comparison | | t | Cohen's d | 95% CI Lower | 95% CI Upper | $p_{holm}$ | Sig |
|---|---|---|---|---|---|---|---|
| DL-DL | M1-DL | 0.20 | 0.04 | -0.57 | 0.66 | 1.00 | |
| | M1-M1 | -4.76 | -0.97 | -1.73 | -0.22 | <0.001 | *** |
| | M1-PMC | -5.61 | -1.15 | -1.95 | -0.34 | <0.001 | *** |
| | PMC-DL | -1.10 | -0.23 | -0.85 | 0.40 | 1.00 | |
| | PMC-PMC | -6.13 | -1.25 | -2.09 | -0.42 | <0.001 | *** |
| M1-DL | M1-M1 | -4.96 | -1.02 | -1.79 | -0.25 | <0.001 | *** |
| | M1-PMC | -5.81 | -1.19 | -2.01 | -0.37 | <0.001 | *** |
| | PMC-DL | -1.31 | -0.27 | -0.90 | 0.36 | 1.00 | |
| | PMC-PMC | -6.33 | -1.30 | -2.15 | -0.44 | <0.001 | *** |
| M1-M1 | M1-PMC | -0.85 | -0.17 | -0.79 | 0.45 | 1.00 | |
| | PMC-DL | 3.66 | 0.75 | 0.04 | 1.45 | 0.002 | ** |
| | PMC-PMC | -1.37 | -0.28 | -0.91 | 0.35 | 1.00 | |
| M1-PMC | PMC-DL | 4.50 | 0.92 | 0.18 | 1.67 | <0.001 | *** |
| | PMC-PMC | -0.52 | -0.11 | -0.72 | 0.51 | 1.00 | |
| PMC-DL | PMC-PMC | -5.02 | -1.03 | -1.80 | -0.26 | <0.001 | *** |

DL represents DLPFC, PMC represents PMC-SMA.

such as direction, speed, and magnitude [51,59,60]. Moreover, given the positive relationship between DLPFC activation and cognitive load during motor task [61], we attribute the lower levels of HbO and FC in DLPFC to the participant's long-term training experience (3.52±1.25 years), which may render the squat movement more automated [62], thereby reducing the cognitive load and DLPFC activation during BFR training.

This study has three limitations. Firstly, our primary focus was on the acute responses of the cerebral cortex during BFR training under different pressure intensities. The effects of long-term BFR training on the cerebral cortex were out of our radar. Secondly, the fNIRS technology used in this study for monitoring brain activity is limited to the cortical surface and cannot detect activity in sub-cortical brain tissue. Lastly, many variables affect the outcome of BFR training, such as subjects' athletic ability, method of compression, material of the compression band, etc. Future researchers could focus on these variables and combine longitudinal study design with other brain imaging technology, like fMRI, to investigate the effects of long-term BFR training on the structure and function of the brain.

## Conclusion

Low-load RT with BFR under different pressure intensities elicits acute responses in the cerebral cortex, and moderate pressure intensity optimally enhances cortical activation. The M1 and PMC-SMA play crucial roles during BFR training through the regulation of activation and FC. In summary, this study provides evidence for the occurrence of regulation of CNS during low-load RT with BFR.

## Supporting information

**S1 Appendix. Sample size estimation.**
(DOCX)

**S2 Appendix. fNIRS data process.**
(DOCX)

## Author Contributions

**Conceptualization:** Binbin Jia, Danyang Li, Wangang Lv.

**Data curation:** Binbin Jia, Chennan Lv.

**Formal analysis:** Binbin Jia, Chennan Lv.

**Investigation:** Binbin Jia.

**Methodology:** Binbin Jia, Chennan Lv, Danyang Li.

**Project administration:** Danyang Li, Wangang Lv.

**Resources:** Danyang Li, Wangang Lv.

**Software:** Binbin Jia.

**Supervision:** Danyang Li, Wangang Lv.

**Validation:** Wangang Lv.

**Visualization:** Binbin Jia, Chennan Lv.

**Writing – original draft:** Binbin Jia, Chennan Lv.

**Writing – review & editing:** Binbin Jia, Chennan Lv, Danyang Li, Wangang Lv.

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
