## [Decision Letter · Decision Letter 0]

29 Jan 2024

PONE-D-23-39697Cerebral cortex activation and functional connectivity during low-load resistance training with blood flow restriction: An fNIRS studyPLOS ONE

Dear Dr. Jia,

Thank you for submitting your manuscript to PLOS ONE. After careful consideration, we feel that it has merit but does not fully meet PLOS ONE’s publication criteria as it currently stands. Therefore, we invite you to submit a revised version of the manuscript that addresses the points raised during the review process.

 There were significant concerns raised by both reviewers, however, I would like to give you the opportunity to address the concerns. Please make sure you put exact p values and ensure that the discussion of findings aligns with the study design of the present work.

We look forward to receiving your revised manuscript.

Kind regards,

Jeremy P Loenneke

Academic Editor

PLOS ONE

Journal Requirements:

"This work was supported by the

Scientific Research Center at Wuhan sports university, China under project number 2022J03."

4. We note that Figure 1 includes an image of a [patient / participant / in the study]. 

Reviewers' comments:

Reviewer's Responses to Questions

**Comments to the Author**

1. Is the manuscript technically sound, and do the data support the conclusions?

Reviewer #1: Partly

Reviewer #2: No

2. Has the statistical analysis been performed appropriately and rigorously? 

Reviewer #1: Yes

Reviewer #2: Yes

3. Have the authors made all data underlying the findings in their manuscript fully available?

Reviewer #1: Yes

Reviewer #2: No

4. Is the manuscript presented in an intelligible fashion and written in standard English?

Reviewer #1: No

Reviewer #2: Yes

5. Review Comments to the Author

Reviewer #1: comments uploaded

Title of Article: Cerebral cortex activation and functional connectivity during low-load resistance training with blood flow restriction: An fNIRS study

General Comments: The authors did a nice job presenting the recent literature as the foundation for the research question utilizing fNIRS to evaluate functional connectivity of different brain regions during resistance training under blood flow restriction. The authors highlight the importance of investigating this phenomenon, as there are many mechanisms associated with blood flow restriction training; it is crucial to also evaluate the neurophysiological mechanisms. However, I feel that the paper needs several major changes in order to strengthen the overall paper. The underlying mechanisms associated with CNS activation and FC regarding resistance training should be highlighted much earlier – in the introduction – to allow for a nice framework for the purpose of the study. The lack of a control group to evaluate differences between BFR / non-BFR training greatly affects the strength of any conclusions to be made regarding adaptations or activity. Furthermore, the absence of a limitations section to address any potential factors associated with the paper are concerning. A more thorough review of the manuscript for grammatical errors and sentence structure is needed, as well as limiting causal language when describing findings. The work presented would be a great first experiment for a future experiment that evaluates the FC between groups, in which more appropriate conclusions could be drawn regarding adaptations; even more so if these adaptations could be evaluated after an intervention study utilizing BFR and its influences on FC and “muscle output” during regular resistance training. Overall, I feel that, with these revisions in mind, the final project will be a solid contribution to the literature. The authors should be commended for all their hard work that was put into this project.

Specific Comments: Specific comments related to my aforementioned general comments are noted below.

Introduction

Line 61: be clear throughout the entire manuscript when discussing resistance training v. resistance exercise – it seems later on it is exclusively written as resistance training (RT), so consider changing this here.

Line 64: consider changing to “investigate potential mechanisms”.

Line 65: with the active verb investigate already earlier in the sentence, consider editing sentence structure to “. . . hormone regulation, and other mechanisms at the cellular and molecular level”.

Line 77: throughout the manuscript, the term adaptation is used to explain the findings of increased cerebral cortex activity due to BFR … with no control group to compare to or increase in some index of performance after the implementation of BFR training, is it suitable to characterize this as an adaptation? Consider justifying this (potential mechanisms associated with increased activity) or explain findings as the increased activity of the cerebral cortex (as stated in the title) and allow for future work to evaluate this as a suitable adaptation due to BFR.

Line 88: consider adding “the” when referencing the prior study (17)

Line 106: the intro discussed 1RM intensities between 20-30%, is there a rationale for choosing 30% for this paper?

Lines 108 – 112: consider editing sentence structure for a better understanding and delivery of second hypothesis.

Methods

Line 125: what neurological and psychological disorders were excluded? Consider being more specific here.

Line 183: consider changing “hypothesis” to “hypotheses” as it seems to be referencing the multiple hypotheses of the paper.

Results:

Consider changing the p values that are reported to = 0.00 to < 0.001.

In Table 1, assuming it is due to rounding, it is noted that the second p value of 0.05 is significant … consider making this clearer or reporting the p value that was less than .05.

Discussion

Line 247: the main finding mentions cortical activation not cortical adaptation – this conclusion seems more relevant to the purpose of this experiment and should match the rest of the manuscript when discussing the purpose.

Line 249-252: as stated above, consider laying out the underlying theoretical framework regarding CNS activation earlier in the introduction to emphasize the importance of the investigation.

Line 253: consider changing “liming” to “limits”.

Line 255: edit sentence structure / grammar – what is it meant by cerebral energy supply? Consider reframing this explanation as “cerebral oxygenation” in line with previous references in the manuscript.

Line 257: the referenced article evaluated the activity during index finger contractions … is this applicable to squat movement and associated brain activity? If so, consider mentioning this paper’s methodology / rationale for cortical activity across different muscular movements.

Line 261: what is it meant by the term “muscle output”? is this meant to be force output? Or muscle activity? Motor unit recruitment? Clearly define what this term means to allow for a better link with findings.

Lines 261 – 265: edit grammar and sentence structure.

Line 288: causal language of “confirming” CNS activity with BFR training. Is there not CNS activity with non-BFR training? Consider elaborating on this finding of augmenting CNS activity with BFR, and what future work could evaluate with this in mind.

Line 293: it is highlighted that there is already high FC during RT without BFR, is this from previous research? If so, which articles should be cited? Moreover, the lack of a control condition to evaluate this outcome limits the overall strength of the study.

Line 301: consider changing to “. . . as the metabolic stress within muscles rises with high pressure”.

Line 302: consider changing “needs to” to “must”.

Line 310: instead of “dominate” use “dominant”.

Lines 312 – 314: edit grammar and sentence structure.

Conclusion

I believe the future work listed would greatly benefit the overall paper / project if a follow-up study was conducted evaluating the differences between a BFR and non-BFR condition. If the goal is to evaluate this effect from BFR as an adaptation, consider an intervention study and looking at pre-post CNS activity – could this be beneficial for overall RT? Populations beyond training populations could benefit as well.

Line 323: consider changing to “neurophysiological mechanisms” as there are more than just one.

Reviewer #2: Thank you for the opportunity to review this manuscript. The author sought to investigate the effects of different pressures on cerebral cortex activation and functional connectivity during the squat exercise. While the design of this study is interesting, I have several concerns about the writing and the claims made by the authors. The authors discuss the adaptation that occur from blood flow restriction training however, this study was acute and therefore cannot make such claims.

Please see the specific comments below.

Lines 65-66 “Surprisingly, even though neural adaptation has been a significant contributor to muscle hypertrophy and strength gain (9–11),”

I am not sure the role that neural adaptations have on hypertrophy. I think it would be helpful if the authors could elaborate more.

Lines 69-71 “In addition, a study has shown that muscle hypertrophy can transfer from muscles exposed to BFR to muscles not exposed to BFR (14).”

I am not sure this study measured hypertrophy. Please see doi: 10.1111/j.1475097X.2010.00949.x

Lines 121-123 “The sample size determination was based on prior-power analysis in G*Power and MPower (32,33). More details about the prior power analysis can be found in Supplementary Materials.”

this is not sufficient to replicate your power calculations. The information in the supplemental materials is also insufficient. Please provide further detail on what variable and that parameters were used to estimate sample size.

I also noticed the subject’s anthropometrics were not provided. Please report their height weight and their max strength. I apologize if I missed it.

For the discussion all talk about the adaptations need to be removed as it is not possible to make such claims from an acute study

6. PLOS authors have the option to publish the peer review history of their article (what does this mean?). If published, this will include your full peer review and any attached files.

Reviewer #1: No

Reviewer #2: No

---

## [Author Response · Author response to Decision Letter 0]

8 Apr 2024

Rebuttal letter

Response: We have made adjustments to the font, font size, line spacing, and first-line indentation in the manuscript to meet the requirements of PLOS ONE style.

2. Please state what role the funders took in the study.  If the funders had no role, please state: "The funders had no role in study design, data collection and analysis, decision to publish, or preparation of the manuscript." If this statement is incorrect you must amend it as needed. 

Response: We have stated the role of the project funder during this study in the Cover Letter. “This work was supported by the Scientific Research Center of Wuhan Sports University, China. Projects included the Young Faculty Research Project and Postdoctoral Research Project. The funder (Binbin Jia) had participated in study design, data collection, and analysis, the decision to publish, as well as preparation of the manuscript.”

3. Please include captions for your Supporting Information files at the end of your manuscript, and update any in-text citations to match accordingly. 

Response: We have added captions (S1 Appendix. Sample Size Estimation & S2 Appendix. FNIRS Data Process) for the Supporting Information files in our manuscript, as well as in-text citations. For instance, “More details about the prior power analysis can be found in the S1 Appendix.” and “The data processing workflow is illustrated in Figure 4, and more details can be found in the S2 Appendix.”

4. The individual(s) or parent(s)/guardian(s) must be informed of the terms of the PLOS open-access (CC-BY) license and provide specific permission for publication of these details under the terms of this license. Please amend the methods section and ethics statement of the manuscript to explicitly state that the patient/participant has provided consent for publication.

Response: We have informed the individuals involved in Figure 1 to provide permissions for publication under the terms of the PLOS open-access (CC-BY), as shown in the following figure. Moreover, we added a statement in the Method section to explicitly state that the participants have provided consent for publication. For example, “The individuals in this photograph have given written informed consent (as outlined in PLOS consent form) to publish these case details.”

Figure 1. Permission from individuals

6.While revising your submission, please upload your figure files to the Preflight Analysis and Conversion Engine (PACE) digital diagnostic tool, https://pacev2.apexcovantage.com/. PACE helps ensure that figures meet PLOS requirements. To use PACE, you must first register as a user. Registration is free. Then, log in and navigate to the UPLOAD tab, where you will find detailed instructions on how to use the tool. If you encounter any issues or have any questions when using PACE, please email PLOS at figures@plos.org. Please note that Supporting Information files do not need this step.

Response: All figures were uploaded to https://pacev2.apexcovantage.com/ to meet PLOS requirements. The images that have been adjusted through PACE have been re-uploaded.

Reviewer #1: General Comments

1. The underlying mechanisms associated with CNS activation and FC regarding resistance training should be highlighted much earlier – in the introduction – to allow for a nice framework for the purpose of the study. 

Response: Great advice. This adjustment allows the introduction part to be more comprehensive and aligns better with the subsequent discussion on the underlying mechanisms. Thus, we have added “it is crucial to provide robust evidence for the characteristics of activity within the central nervous system (CNS), such as activation and functional connectivity (FC) of the cerebral cortex. Those indices not only serve as excellent windows for exploring the response pattern of the cerebral cortex but have also been confirmed to undergo adaptive changes with resistance training[19]. ” “The first hypothesis is that pressure intensity regulates cortical activation. Furthermore, we infer activation and FC strength would enhance with pressure intensity to improve muscle force output based on previous studies[29,30,17]. This is because the increasing metabolic stress under BFR restricts the capacity of muscle[31,6], and the CNS can compensate for the muscle force loss under BFR by enhancing the recruitment of motor units and improving the frequency of neural impulse discharge[32].”

2. The lack of a control group to evaluate differences between BFR / non-BFR training greatly affects the strength of any conclusions to be made regarding adaptations or activity.

Response: Many thanks for this thoughtful comment. Without a doubt, a control group would strengthen the conclusions drawn from our study. However, the individual difference is something to consider when the between-subject design is employed. This is also the main reason we took advantage of within-subject design for its better control of individual differences and higher statistical power. Well, we actually designed a control condition in our experiment, which is the 0 mm Hg pressure intensity for comparison between BFR and non-BFR training. To be honest, the RCT design would be our first choice. However, we had to make some trade-offs in the experimental design due to the limited research budget. Anyway, we have revised the relevant statements in the Introduction part to make our control condition more intuitive and understandable. Like, “Specifically, we intend to combine the 30% 1RM squat exercise with BFR under different pressure intensities (150 mmHg, 250 mmHg, 350 mmHg, and 0 mmHg or non-BFR as a control condition) to investigate the cortical activation and FC in M1, PMC-SMA, DLPFC via fNIRS. ”

3. The absence of a limitations section to address any potential factors associated with the paper is concerning.

Response: I totally agree. We did consider the limitations of our study in the first draft. However, the limitation section is not a mandatory component for Plos One. Here we provide limitations like “This study has three limitations. Firstly, our primary focus was on the acute responses of the cerebral cortex during BFR training under different pressure intensities. The effects of long-term BFR training on the cerebral cortex were out of our radar. Secondly, the fNIRS technology used in this study for monitoring brain activity is limited to the cortical surface and cannot detect activity in sub-cortical brain tissue. Lastly, many variables affect the outcome of BFR training, such as subjects’ athletic ability, method of compression, material of the compression band, etc. Future researchers could focus on these variables and combine longitudinal study design with other brain imaging technology, like fMRI, to investigate the effects of long-term BFR training on the structure and function of the brain.” Undoubtedly, this section would provide valuable guidance for researchers in the field of BFR training. We have already added this part to the current draft, however, whether it will appear in the final article requires the journal’s approval.

4. A more thorough review of the manuscript for grammatical errors and sentence structure is needed, as well as limiting causal language when describing findings.

Response: Thanks for this suggestion. The grammar and sentence structure of this draft definitely needs improvement. To do so, we have incorporated a proficient English teacher for comprehensive editing. Additionally, we have adjusted the statements regarding causal inference in the text to better correspond with our research findings.

Reviewer #1: Specific Comments

1. Line 61: be clear throughout the entire manuscript when discussing resistance training v. resistance exercise – it seems later on it is exclusively written as resistance training (RT), so consider changing this here.

Response: Corrected. “While BFR has combined with various types of exercise, research indicates that the most substantial muscular gains come with resistance training (RT) under 20%-40% of the 1 repetition maximum (1RM) or maximum voluntary contraction (MVC) [5].”

2. Line 64: consider changing to “investigate potential mechanisms”.

Response: Corrected. “With the increasing popularity of BFR in the training domain, many researchers have started to investigate its potential mechanisms”

3. Line 65: with the active verb investigate already earlier in the sentence, consider editing the sentence structure to “. . . hormone regulation, and other mechanisms at the cellular and molecular level”.

Response: Corrected. The sentence has been edited to “many researchers have started to investigate its potential mechanisms, such as metabolic stress, cellular swelling, hormone regulation, and other mechanisms at the cellular and molecular level.”

4. Line 77: throughout the manuscript, the term adaptation is used to explain the findings of increased cerebral cortex activity due to BFR … with no control group to compare to or increase in some index of performance after the implementation of BFR training, is it suitable to characterize this as an adaptation? Consider justifying this (potential mechanisms associated with increased activity) or explain findings as the increased activity of the cerebral cortex (as stated in the title) and allow for future work to evaluate this as a suitable adaptation due to BFR.

Response: Thanks for this advice. As you mentioned, the HbO change of the cerebral cortex during our experiment only represents cortical activity intensity to some extent and cannot fully correspond to the description of neural adaptation. Therefore, we have decided to follow your suggestion and adjust the relevant descriptions of adaptation in the text to “characteristics of cortical activity” “cortical responses” or “cortical activation”.

5. Line 88: consider adding “the” when referencing the prior study (17)

Response: Corrected. “ there is a possibility that the increases in cortical activation from the prior study [18] ”

6. Line 106: the intro discussed 1RM intensities between 20-30%, is there a rationale for choosing 30% for this paper?

Response: The 30%1RM load was selected due to its popularity. Based on the review (Blood Flow Restricted Exercise and Discomfort: A Review) from Spitz et, al. (2020), the 30% 1RM was the most used load for acute BFR+RE training studies (see Figure below). 

7. Lines 108 – 112: consider editing sentence structure for a better understanding and delivery of the second hypothesis.

Response: Thanks for your comments. We have changed the sentence to make it more specific and clear. “The first hypothesis is that pressure intensity affects cortical response. Furthermore, we infer activation and FC strength would enhance with pressure intensity to improve muscle force output [32,33,19]. This is because the increasing metabolic stress with BFR restricts the capacity of muscle [34,6], and the CNS can compensate for the muscle force loss under BFR by enhancing the recruitment of motor units and improving the frequency of neural impulse discharge[35], resulting in higher activation. Additionally, considering the changes in cortical HbO during BFR training may result from altered blood distribution and increased cerebral blood flow [36,37], rather than active regulation of the CNS, we propose the second hypothesis that the influence of pressure intensity on cortical activation is moderated by the regions of interest (ROI). ”

8. Line 125: what neurological and psychological disorders were excluded? Consider being more specific here.

Response: Corrected. Those disorders include, but are not limited to Depression, Autism, Mania, Schizophrenia, Epilepsy, and Stroke. We have added those disorders for more specific expression in the Method section. “Exclusion criteria included neurological or psychological disorders (Depression, Autism, Mania, Schizophrenia, Epilepsy, Stroke, etc.)”

9. Line 183: consider changing “hypothesis” to “hypotheses” as it seems to be referencing the multiple hypotheses of the paper.

Response: Corrected. “Repeated measures analysis of variance (RMANOVA) was employed to test our hypotheses.”

10. Consider changing the p values that are reported to = 0.00 to < 0.001.

In Table 1, assuming it is due to rounding, it is noted that the second p-value of 0.05 is significant … consider making this clearer or reporting the p-value that was less than .05. 

Response: Thanks for your thorough review. We have carefully implemented the necessary corrections to the relevant content according to your suggestion for more precise expression, see Table 1. In addition, the statistical results of our research were uploaded to the open-access website https://figshare.com/s/12cb30992566b4dc1b18 for those who need more specific results (like p-value without rounding). See the Data analysis and statistics part, “The raw data of fNIRS, R code and statistical results from JASP can be found at https:// osf.io/5r2qp/.”

Table 1 Post Hoc Comparisons - Pressure Intensity (HbOmean)

 95% CI 

Comparison t d Lower Upper pholm Sig

0mmHg 150mmHg -0.62 -0.09 -0.50 0.32 0.54 

 250mmHg -2.49 -0.36 -0.80 0.07 0.04 *

 350mmHg 2.78 0.41 -0.03 0.85 0.03 *

150mmHg 250mmHg -1.88 -0.27 -0.70 0.15 0.13 

 350mmHg 3.40 0.50 0.04 0.95 0.006 **

250mmHg 350mmHg 5.28 0.77 0.25 1.29 <0.001 ***

11. Line 247: the main finding mentions cortical activation, not cortical adaptation – this conclusion seems more relevant to the purpose of this experiment and should match the rest of the manuscript when discussing the purpose.

Response: Agreed. We have modified the relevant statements regarding cortical adaptation to better align with the research objectives by using the term “cortical regulation” or “cortical activity”. For example “This research confirms the occurrence of the cortical regulation during BFR training.”

12. Line 249-252: as stated above, consider laying out the underlying theoretical framework regarding CNS activation earlier in the introduction to emphasize the importance of the investigation.

Response: Thanks for your kind reminder. We have already integrated your recommendation into the introduction section as we responded to General Comment 1.

13.Line 253: consider changing “liming” to “limits”.

Response: Corrected. “which limits the muscle capacity”

14. Line 255: edit sentence structure/grammar – what is it meant by cerebral energy supply? Consider reframing this explanation as “cerebral oxygenation” in line with previous references in the manuscript.

Response: Following your advice, we have changed the sentence structure to make it more clear and sound, “Consequently, there is an elevation in cortical HbO, ensuring cerebral energy supply and subsequently leading to increased cortical activity during BFR training.”

14.Line 257: the referenced article evaluated the activity during index finger contractions … is this applicable to squat movement and associated brain activity? If so, consider mentioning this paper’s methodology/rationale for cortical activity across different muscular movements.

Response: This suggestion is quite valid. However, due to the limited neuro-imaging studies on brain activity during whole-body resistance training, we had to consider similar research to assist in inferring changes in cortical activity during squat with BFR in our study. This is mainly based on the assumption that the central nervous system's regulation of muscle force output should be consistent, whether the exercise involves major muscle groups or minor muscle groups. Additionally, we have augmented the reliability of our inferences by including neuro-imaging studies that employ squat training and fNIRS similar to those used in our research as references. (Hemodynamic Response Alterations in Sensorimotor Areas as a Function of Barbell Load Levels during Squatting: An fNIRS Study, 2017). “Furthermore, this heightened activity has been consistently ass

---

## [Decision Letter · Decision Letter 1]

6 May 2024

Cerebral cortex activation and functional connectivity during low-load resistance training with blood flow restriction: An fNIRS study

PONE-D-23-39697R1

Dear Dr. Jia,

We’re pleased to inform you that your manuscript has been judged scientifically suitable for publication and will be formally accepted for publication once it meets all outstanding technical requirements.

Kind regards,

Jeremy P Loenneke

Academic Editor

PLOS ONE

Additional Editor Comments (optional):

The only remaining change is to change "between group" to "between condition" since this is a within participant design. That can probably be made in the editing stage.

Reviewers' comments:

Reviewer's Responses to Questions

**Comments to the Author**

1. If the authors have adequately addressed your comments raised in a previous round of review and you feel that this manuscript is now acceptable for publication, you may indicate that here to bypass the “Comments to the Author” section, enter your conflict of interest statement in the “Confidential to Editor” section, and submit your "Accept" recommendation.

Reviewer #1: All comments have been addressed

Reviewer #2: All comments have been addressed

2. Is the manuscript technically sound, and do the data support the conclusions?

Reviewer #1: Yes

Reviewer #2: Yes

3. Has the statistical analysis been performed appropriately and rigorously? 

Reviewer #1: Yes

Reviewer #2: Yes

4. Have the authors made all data underlying the findings in their manuscript fully available?

Reviewer #1: Yes

Reviewer #2: Yes

5. Is the manuscript presented in an intelligible fashion and written in standard English?

Reviewer #1: Yes

Reviewer #2: Yes

6. Review Comments to the Author

Reviewer #1: (No Response)

Reviewer #2: That authors have addressed my concerns.

I have one more concern. I must apologize for not catching this earlier. But the authors should change between group to between conditions as this was a within subject design.

7. PLOS authors have the option to publish the peer review history of their article (what does this mean?). If published, this will include your full peer review and any attached files.

Reviewer #1: No

Reviewer #2: No

---

## [Editor Report · Acceptance letter]

14 May 2024

PONE-D-23-39697R1 

PLOS ONE

Dear Dr. Jia, 

I'm pleased to inform you that your manuscript has been deemed suitable for publication in PLOS ONE. Congratulations! Your manuscript is now being handed over to our production team.

Kind regards, 

on behalf of

Dr. Jeremy P Loenneke 

Academic Editor

PLOS ONE